# ARTS, an AR Tourism System, for the Integration of 3D Scanning and Smartphone AR in Cultural Heritage Tourism and Pedagogy

**DOI:** 10.3390/s19173725

**Published:** 2019-08-28

**Authors:** Naai-Jung Shih, Pei-Huang Diao, Yi Chen

**Affiliations:** Department of Architecture, National Taiwan University of Science and Technology, 43, Section 4, Keelung Road, Taipei 106, Taiwan

**Keywords:** augmented reality, LIDAR, point cloud, cultural heritage, 3D reconstruction, pedagogy, Lukang, Shih Yih Hall

## Abstract

Interactions between cultural heritage, tourism, and pedagogy deserve investigation in an as-built environment under a macro- or micro-perspective of urban fabric. The heritage site of Shih Yih Hall, Lukang, was explored. An Augmented Reality Tourism System (ARTS) was developed on a smartphone-based platform for a novel application scenario using 3D scans converted from a point cloud to a portable interaction size. ARTS comprises a real-time environment viewing module, a space-switching module, and an Augmented Reality (AR) guide graphic module. The system facilitates scenario initiations, projection and superimposition, annotation, and interface customization, with software tools developed using ARKit^®^ on the iPhone XS Max^®^. The three-way interaction between urban fabric, cultural heritage tourism, and pedagogy was made possible through background block-outs and an additive or selective display. The illustration of the full-scale experience of the smartphone app was made feasible for co-relating the cultural dependence of urban fabric on tourism. The great fidelity of 3D scans and AR scenes act as a pedagogical aid for students or tourists. A Post-Study System Usability Questionnaire (PSSUQ) evaluation verified the usefulness of ARTS.

## 1. Introduction

Although urban fabric, cultural heritage (CH) tourism, and pedagogy are usually explored in different study domains, similar concerns and shared subjects exist. A fabric is created through a chronological progress of government promotions, local commercial efforts, and/or private residential renovations. All accumulated efforts facilitate the evolution of tourism and activate cultural transfer. To understand local identity and culture, knowledge tourism usually explores the developing lines of chronological efforts from physical on-site experiences to annotate the evolving progress. Tourism can fulfill the pedagogical needs of architectural education in history, conservation, and/or urban planning. While the fabric constitutes a collaged design or installation of artifacts focusing on an as-built stage of development, well-planned pedagogy will support domain-specific learning, such as architecture history, Taiwanese architecture, and/or urban design. Tourism leads to interactions between urban fabric and roles in terms of the links, causes, or meanings of the urban fabric. The conveyed roles include tourists, students, cultural workers, planners, architects, residents, or administrators, under varied levels of exploration, verification, or entertainment.

Previous studies have identified the close inter-relationship between cultural heritage tourism, entertainment, and pedagogy. Heritage tourism attracts visitors, as well as fulfills the social, cultural, and recreational aspirations of the local people [1]. Maps are carried around to enrich their trips. The engaged tourist behaviors constitute walking, reading maps, visiting attractions, shopping, and resting [2]. Visitors learn about a site from a combination of relaxation and educational interests [3]. The promotion of tourism increases opportunities to explore traditional culture. Most local residents support tourism for the economic, cultural, and societal benefits that it imparts [4]. The positive attitude of tourists and local residents can also stimulate the involvement of investors and influence the planning and management strategy of local environments [5].

Interactions between the urban fabric, tourism, and pedagogy deserve investigation regarding whether a detailed exemplification exists in an as-built environment at the macro- or micro-perspective of operation. The fabric is shaped by a collaborative effort between users and buildings which are designed or owned by different parties. Issues exist concerning whether or not the interpretation of urban fabric can be extended beyond documents and survey drawings, such as maps, satellite images, or sensor-based data to different levels of accuracy or detail during interactions. It remains to be determined if interactions should occur with the features presented in Boolean-like fabric operations starting from micro-expressions, skyline compositions, and section overlays, using the details presented on second skins and new construction. The result of operations which are a part of pedagogy efforts should feed-back to cultural tourism with experience from older and more established elements.

## 2. Purposes

Interactions made to 3D scan data should be explored to a level that is feasible for pedagogy or tourism use. To achieve this, an augmented reality (AR) tourism system, or ARTS, was developed to integrate smartphone AR and a 3D scanning model to facilitate the representation of urban fabric in a dependent and co-existing relationship with cultural elements. An old street was purposely selected and 3D-scanned for the exemplification of ARTS. Its importance in historical development and the combinative nature of local characteristics enabled the development of a markerless AR system for a superior tourism experience using 3D scanned data.

## 3. Related work

The tourism industry has been greatly influenced by information technology, the Internet, and the telecommunication industry for years [6,7]. Related studies can be found that focus on eTourism or mobile tourism [8,9,10]. Virtual reality (VR) promotes eTourism by overcoming a lack of resources or safety concerns with unique visual experiences [11]. AR offers a number of advantages in tourism. For example, a tourist experience can be created without a guide [12]. Images or objects can be superimposed to introduce historical buildings or events. Compared to online sources or travel guides, a large amount of relevant tourist information can be represented in different and enjoyable forms [13,14,15].

Digital technology has significantly influenced the evolution of cultural heritage dissemination, leading to the advent of new research branches [16,17]. AR, which combines real and virtual objects in an environment with real-time operation [18,19], constitutes a source of innovation to co-create enhanced CH experiences [20,21,22]. This technology has tremendous potential for the promotion and preservation of CH [23]. The combination of physical objects with virtual information has stimulated the interest of students to physically explore archaeological sites [24].

AR applications have been developed in a variety of cultural heritage tourism sites and scenarios. For example, Google Glass^®^ was applied in a gallery to enhance visitor experience by overlaying information with paintings [25]. A Baroque vault, in existence between 1682 and 2006, was visualized through its augmentation to an existing central Gothic vault for an interior AR-cathedral exploration [26]. For a large exterior site, projection-based AR was applied on an existing building facade in Disney Parks [27].

The study of a heritage site usually requires the creation of a 3D model for preservation or tourism purposes. The model can be used for documentation, measurement, or illustration. The recorded contents not only illustrate the open space, but also the urban fabric that is made of buildings, culture entities, or the landscape. Photogrammetry technology has been widely applied in unmanned aerial vehicles (UAVs) or unmanned aircraft systems (UASs) [28,29]. Four-dimensional dynamic scene reconstruction has been developed as an efficient modeling approach using videos or photos as input resources of images [30]. For the capture of detailed models, 3D scanning can create objects of the original scale (in 1:1) to fulfill measurement needs in architectural professional practice. An automatic scan process can be made to a scene up to 360 × 300 degrees of an environment in a range of 100 m. The automatic registration of multiple scans can be made, no matter whether it is a large site or a long alley. A point cloud model has been widely used for construction documents.

Tourism is closely related to the design and planning of urban development. Three-dimensional city models, which were created by 3D scans and photogrammetry, have been displayed using AR for promotion purposes [31]. Virtual static and dynamic objects have also been augmented for the visualization of proposed modifications of urban environments [32]. A mobile augmented reality tool has been applied to collect crowd data or decisions to be referenced by urban design and planning [33].

Problems occur with AR or VR under different technology limits. Although marker-based AR and markerless AR constitute two major AR types [34], the former is often applied in tourism-related research. Considering the difficulty and aesthetic concerns involved in arranging and pasting markers on building surfaces, markerless AR is a more suitable approach for diversified scenes [35,36]. Indeed, the perception or detection of interferences to a path can be missing in real environments [37].

## 4. The Macro Perspective in History and Urban Fabric

Different fabric relationships lead to a reconceptualization of urban data that can be utilized to support the representation of detailed heritage configuration. The data can be converted to desired forms as needed. This is particularly the case when studies include as-built scan data for a full-scale description of an environment. Urban fabric stands for building enclosures that define open spaces. The retrieval of urban fabric usually comes from geographic maps in 2D. Recent studies have emphasized second contours (or second skins), which contribute a new, remodeled, and yet flexible existence of installations to the definition of urban fabric [38]. Two-dimensional maps can no longer meet the demand in terms of the geometries or surface attributes of descriptions. An effective and efficient manner of 3D scanning has been applied to retrieve related data. The level of detail of the as-built data enables a unique perspective in defining fabric types, relationships, and operations that can be applied to facilitate studies in eTourism and architectural pedagogy.

Lukang, a heritage town in Taiwan, has a unique architectural style and urban fabric from the earliest stages of its development (1683–1777), development peak (1784–1836), recession (1851–1888), degeneration (1895–1943), recovery (1945–1987), and current development phase (1988–now). It is located on the west coast of Taiwan and was formerly the second-largest harbor between the 17th and 19th centuries. Its competitive advantage disappeared after port silting and the rejection of railroad deployment [39,40]. The slowdown of local development and modernization eventually prevented the city’s over-expansion and the potential deconstruction of historical heritages.

The distinguishing characteristics in its landscape, humanity, and styles, however, have been well-preserved, making it one of the most important sightseeing locations in Taiwan. A large number of cultural relics exist around the Zhongshan Road, Zhongzheng Road, and peripheral regions, which are also known as old streets. A Heritage Preservation District was established by past government promotion strategies. The old streets have developed into a unique block-based commercial pattern of cultural industry. For example, one linear old alley has connected historical blocks and created a rich tapestry of food, handcrafts, boarding, religions, and open spaces. The tourist routes also vary by subject. Three-dimensional scans have been made of a famous old street over approximately 2500 m (Figure 1).

## 5. The Micro Perspective of Sections

In addition to the presentation of an entire point cloud model, one of the most important spatial analysis approaches in architecture is to create sections to illustrate the vertical proportion of open space in an alley. The sections not only present elevated gestures and artifacts on building enclosures, but also correlate visual experience with elevations at a human scale. A cross-section of neighborhood buildings is a perfect illustration of diversified styles, front access, recess levels, and building facilities. In other words, alley or street sections are perfect for second contour or skin illustration.

The individual setback of a building front usually merges with adjacent spaces on the neighboring buildings to form a unique open space pattern of the street. Slices of alley help to isolate a particular part of the space for profile inspection. A section is a traditional architectural drawing type which is usually used to illustrate the spatial configuration along different altitudes. A section is usually made of a drawing with no thickness. Since a very thin layer of the point cloud may not have enough points to define the edge of a wall boundary, this procedure was modified in this study with a slightly thicker cut for a much clearer illustration of the profile. Sections were applied in the AR system, so a user can (1) move around or even align with a wall to tell the variations of the alley space by setbacks, (2) display sections selectively to study the most different ones, or (3) select all sections and deselect them one by one from near to distant (or vice versa) as a recollection of sequential views.

The sections are not just created at a specific location. Since an old street is a famous local sightseeing location under linear browsing and movement patterns of tourists, sections should be created at intervals to represent characteristics at every sector. In order to correlate an individual section distributed along a street axis visually, a straightforward way was developed to overlay a number of sections together with alignment to the street’s central line to show the varied back-and-forth level of distance to the shared open space.

The sections were sliced from street scans using CloudCompare^®^ at defined intervals with predefined thicknesses. An axis was drawn along the alley centerline (Figure 2). Overlaid sections, which occurred every 200 m (Figure 3), signified an alley’s profile. The sections correlated urban and cultural elements over a range of distances into a short range for a side-by-side comparison of openness, building styles, fabric variations, temporal installations, and landscape. Reading the sections from one end to the other of an alley reveals the intensity of development, change of urban zoning types (e.g., from commercial to residence, or vice versa), local adoption of building codes, definition of private property, etc. All of the illustrations with structural and visual details were made possible by using 3D scan point cloud models, as a type of chronological record for future reference of changes or urban remodeling.

The top left section overlay illustrates contrasting building heights in distinctive styles. Ironically, the contrast was softened by the landscape as a connecting interface that transfers from the past in a low-rise to the left to the relatively newly built buildings to the right with less of a style statement. The styles or constructions reveal an occurring transformational process in its midway or completely finished status, with some aspects still remaining as a memory from the past.

This contrasting image can hardly be defined by a single instance or a deployment promoted by a specific government strategy. The result presents a set of co-existing developing stages that are scattered, on the one hand, in such a short linear distribution of geographic range, and on the other hand, over such a long period of time, involving several generations of local residents.

## 6. Involved Roles

Pedagogical experience is made by co-relating similar entities at points of interest in order to compare, annotate, or extend the scope of vision. Concerning tourism, annotating identity is achieved by a comparison of similar, yet different, styles in order to enhance the student’s comprehension of location-based cultural identity. Learning is attained at the scale of a building, alley, or street block to emphasize the gray areas or highlight specific locations. Learning is also made from approaching views at the building front or near the back entrance. The influences of the collaborative environment include creations, enforced building or urban codes, building life cycle developments, or alley planning strategies. In these settings, the level of pedagogical experience is highly influenced by subjects integrating experience from the past with present perceptions.

### 6.1. Roles

Involvement in tourism should associate the inspection with people, culture, religion, urban spaces, building structure, economy, social issues, etc. The involvement of a tourist or an architecture student in tourism can be simplified into a three-way association with space and pedagogy. The roles involved in the tests of physical fabric entity manipulation are local residents, tourists, cultural workers, and chiefs of villages. The concerns of local residents comprise local identity, privacy invasion, separated private entrance, future development, or government promotion. Additional concerns may be extended to the conflict between conservation and basic living functions, and/or the separation of old property rights and configuration in narrow lots without sufficient access to each. Segregation between tourism and private living boundaries, businesses, and residences is also a major concern. Tourist-related experiences include culture, food, religion, and pedagogy with information, illustration, or a comparison of similar, yet divergent, styles of subjects. For cultural workers, the points of interest include styles, materials, construction methods, historical evolution, humanity, history, and culture-related aspects. Chiefs of villages or neighborhood managers are concerned about local identity, promotion, resident involvement, environmental collaboration, and the delivery and monitoring of government policy.

### 6.2. Roles Exemplifications

The target AR exemplifications are conducted based on the three-way interaction between urban fabric, tourism, and pedagogy. The connection, cause, and meaning of a fabric’s historical background verification comprise government promotion stages, local commercial efforts, individual house renovations, and cultural transfer from old history to new business types. Knowledge tourism’s annotations includes two types—i.e., satellite image overlays and the on-site detection of interfaces occurring between stages of construction—in order to identify their influence on current environmental settings. The changes can be obviously seen in shifts of pavement patterns, construction materials, or facade styles. The development of local commercial efforts also supports religious tourism and simultaneously maintains communication with residents.

The selective displays of single and overlaid sections contrast the differences in fabric, proportions, or silhouettes with a clear illustration of relative locations. A user selects a part of the display according to his or her intention or by following a tutor’s instruction. The location of an overlaid section can be prearranged at the middle or either end of an alley in order to draw attention to specific location-based changes.

The interaction creates several types of scenarios involving local identity or unexpected fabric occurrence, such as approaching views, conflict construction, skylines, and details. The conflict constructions, which were added with different materials and construction methods such as rain shades, appear abruptly as a temporary fabric in a dissimilar style or vocabulary. The micro-expression or micro-fabric is also explored or exemplified through such entities as motorcycles, billboards, canopies, seasonal signage, brick patterns, family originality, and plants in a dynamic or hierarchical co-existing layout.

## 7. AR-Related Tasks and Features

AR representation is a pedagogical experiment of approaching views with a selective display sequence or filtering method from either direction. The display illustrates the chronological influence under the impact of government promotions, local commercial efforts, or individual house renovations. The objects of interest are displayed or hidden purposely with scene partitioning and scenarios, as a result of interaction which uses layers of app interfaces to support the application or evaluation among different roles.

AR expressions are segmented 3D polygons which are transferred from 3D scans. Casually unexpected, yet important, signs of daily life (e.g., a motorcycle in front of a door) and 3D human scale (e.g., proportion of a human figure in a space) are retrieved in order to achieve a more realistic interactive experience of an alley in remote tourism. The vertical urban fabric is represented by overlaying 3D polygon models and 3D illustration models with a semi-transparent polygon of section outlines or semi-opaque polygon to block out conflict construction.

The framework of the app interface is incorporated with groupings or layers of button sets, which allow a user to work with building parts from the left or signs of linear notational frames from the right. Switching to the next layer can also be selected to simulate the on-site environment from a remote location. The test can be conducted in a remote site with or without a synthesized background, such as the sky, or an alternate fabric appearance, such as high-rise buildings.

The main AR-related tasks and features of this app include scenario initiations, scene partitioning, annotations, and interface customization, with the support of software tools for selective display.

Scenario initiations: While tests are conducted by different roles, the objects of interest are displayed selectively to illustrate the potential influence on the alley. The influence may come from promotion from government planning strategies, concerns of local residents, interactions between resident needs and tourism, or simply approaching images from either direction (Figure 4). Each scenario has a specific arrangement of buttons for superimposed display or hide-out.Scene partitioning: Scanned alley scenes are partitioned into three segments: Far, middle, and near ranges. The ground portions of the alley are separated from the top ones, which are the second floor and above, to distinguish the differences between conflict construction and old residences.Annotation
▪Annotation frames: Using buttons to the right, notations can be made to the AR scenes in texts, skylines, section outlines, second contours, or blocking boxes for the display of selective or enhanced contents.▪Additive geometry annotation: The interpretation of notations is represented in 3D geometries that are selectively superimposed on a 3D mesh model (Figure 5). The geometries are segregated by distance and outline overlay. The display list and hierarchy can be managed in order to meet pedagogical needs. For example, a test was conducted by separating an alley approach view into three parts according to distance, along with sections originally created from CloudCompare^®^.▪Collaborative annotation schemes: Notations are collaborated through different kinds of software and hardware for point cloud input, analysis, and decimation. The schemes are determined by locations, roles, or tasks.
Interface customization: This customized system interface allows switching between panels, buttons, or themes to enable pedagogical applications in role-based surveys/investigations.Software tools for selective display: The display leads to the operations as follows:
▪Selective display or overlay: The contrast between old elements and new (or conflict) construction is shown (Figure 6).▪Contrast illustration: Urban context displays with and without points of interest.


## 8. ARTS

A system or an app is a reminiscent representation of the real world. On the one hand, its scope has to be large enough to cover the macro perspective of the history and urban fabric; on the other hand, the manipulated element has to be small enough in the micro perspective to construct the variations. The heritage site of Shih Yih Hall, Lukang was selected based on its unique cultural background and rich neighborhood.

As with the AR-related tasks and features mentioned in the previous section, the development of ARTS was planned through three stages of testing. The first one, which was a marker-based AR system, was applied to 3D models of Pingxi Old Street. AR-media^TM^ was applied and 3D models were created by photogrammetry technology utilizing UAV images. Three-dimensional scenic and local iconic models were displayed interactively upon the scan of a traditional map [31]. The second stage, which was a markerless AR system (Figure 7), was applied to the 3D as-built models of Yingge Old Street using ARKit^®^ and models created by a point cloud of 3D scans. ARTS was built with expanded options in selecting scenes and graphic annotations of sections.

The use of photogrammetry in the workflow for the 3D model creation process was only used at the first stage of the three testing stages in Pingxi Old Street. This was a different site from the one discussed in this paper. The first stage experience was made by combining the digital elevation model (DEM) created from UAV images and the point cloud model from the 3D scanner. Later, the DEM was used for integration with the missing areas which were impossible to collect from the ground level. However, the level of detail in DEM was not quite at as well-defined a level as a point cloud model could create. The region—for example, the corner that connects exterior walls and roof—did not fit well; in other words, seamless fit did not occur where the two boundaries (DEM and the point cloud model) were connected. No DEM model was created at the second and the third stage, since the emphasis was on the walking experience along a representative old street on the ground level.

### 8.1. System Development

ARTS comprises of three modules (Figure 8): (1) A real-time environment viewing module; (2) a space-switching module; and (3) an AR guide graphic module. The first module provides real-time environment inspection with simultaneous localization and mapping (SLAM) enabled. The second module switches the display between a series of spaces, building components, and 3D annotations for different tourism or pedagogical experiences. The third module provides an interface between the user and the 3D mesh model and a point cloud model database.

AR SDK was built on ARKit^®^ and the iPhone XS Max^®^. ARTS was developed in six main procedures (Figure 9). All of the models were imported by Unity^®^ and created an AR scene (Figure 10). Six modules were coded in Microsoft Visual Studio^®^, and an Xcode^®^ project was created for the final app.

### 8.2. User Interface

The ARTS user interface can be seen in Figure 11, in which buttons are located on both sides of the screen with a “hide” option. The left buttons switch the display of streets and buildings in a 1:1 scale. The right buttons switch the annotation displays of the section line (SL) and conflict construction. Buttons appear in a gray color when activated. All of the sub-functions can be hidden for an unobstructed screen display.

ARTS can be applied on-site or at a remote location. The 1:1 3D alley model helps the user to experience a space with higher fidelity and facilitates urban inspection using an as-built scene from a remote location. The annotation model is for on-site spatial information display associated with traditional graphic drawings for the comprehension of spatial relations, even those beyond the specific scene. Operation photographs and interface screen shots can be seen in Figure 10.

The images perceived in the app were displayed in a relatively correct location to a user’s height. The 3D models or sections were properly located, with the hand-held height defined at the initial operational stage being the same as the height in the AR model. The model is in actual scale and so is the user. Although the height of each user may vary, the initial display is located at about the same height as eye level above the ground. Relative movement would not change the size of the model. As a result, there is no need to specify the scale in the AR scene. The way a user walks in a real site is approximated.

The representation and visualization is divided into two categories for the approaching views using six connected parts (Figure 4) and the profile views use overlaid sections of the 11 separated or joined slices (Figure 10). The app includes three major parts of 3D models: The 10 m range of the alley around Shih Yih Hall, a 20 m range of the alley around Shih Yih Hall in 11 separated sections, and the combined 11 sections. The first part was divided into three segments which are located before, after, and containing the Shih Yih Hall to experience approaching views from either direction. Each segment is again divided by the original design and the newly added constructions, such as a weather shelter, in six smaller parts. Upon the selection, the weather shelter can be hidden to reveal the original configuration of the hall. The 11 slices were used to illustrate the alley profile by following a tourist’s walking speed in a separated or accumulated manner. Both layouts were managed for the comprehension of the variations occurring along the alley. Detailed inspection of individual profiles can be made by selection. The joined sections create an accumulated effect as a characteristic conclusion within this 20 m range.

AR browsing at night is also feasible. Combining AR and a 3D model made of night scans can constitute a captivating experience. In addition, some scenes are better shown in the evening or in a room with the lights turned off. A dramatic browsing experience can also be created even in the daytime. These aspects enable increased browsing diversity in the classroom.

The systematic structure of ARTS can be seen in Figure 12, in which “Remote” and “On-site” modes can be selected. The former offers options including “Combine with reality” (Figure 13 and Figure 14) and “Full virtual” to incorporate or compare with the landscape, architecture, or street pavement. “Full virtual” can be utilized to illustrate an entire building with section lines. “On-site use” applies virtual objects as annotations to indicate conflict constructions and section lines.

### 8.3. The Data Work Flow and Scan Data Conversion

The building and environment were scanned into a 3D point cloud model (10 GB), using the Faro^®^ Focus 3D scanner. Due to the limited storage space of a mobile phone, an alley approximately 20 m next to Shih Yih Hall was selected. Increasing the polygon number causes a delay in the screen display with flickering. Decimation was made by the defined polygon count or percentage. The cloud model was converted to a mesh model in OBJ and FBX format using Geomagic Studio^®^ and the Autodesk FBX Converter^®^, respectively. The original point clouds were converted from points, polygons, decimated polygons, OBJ format, and FBX format, to AR files in several stages.

The entire model was made around Shih Yih Hall. For display efficiency, the original mesh model of 301.4 MB was divided into 18 parts in six adjacent facades in a top-and-bottom arrangement, 11 facade slices along the street axis, and one overlaid facade of the 11 slices. The first larger model was divided into six parts. Each part was approximately 400,000–1,000,000 polygons, comprising about 32% of the original polygon numbers. The six sizes of all six parts were reduced from 668.7 MB in PLY format, to 324 MB in OBJ format, to 159.1 MB in FBX format, constituting 22.5% of the original size. The 11 slices, which were made of 60 million points amounting to about 1 GB, were used to illustrate the alley. The data were decimated and converted to about 2.2 million polygons in 452 MB, prior to the final size of 1,000,000 polygons being achieved. The 11 parts were also combined. Both separated and combined models totaled 142.3 MB.

An iPhone XS Max^®^, which was equipped with an A12 CPU with 4 GB RAM and 64 GB internal storage, was used. The final size of the app is 321.6 MB.

## 9. System Evaluation

The purpose of this AR study is to test the feasibility and satisfaction of using ARTS. In the evaluation of satisfaction, the post-study system usability questionnaire (PSSUQ) was used in four dimensions, with a total of 16 items, including 1–16 items for “overall average”, 1–6 items for “system usefulness”, 7–12 items for “information quality”, and 13–15 items for “interface quality.” The questionnaire items were scored on a Likert-type seven-point scale [41], where 1, 2, 3, 4, 5, 6, and 7 represented “strongly agree”, “agree”, “somewhat agree”, “neutral”, “somewhat disagree”, “disagree”, and “strongly disagree”, respectively. Essentially, the lower the rating is, the higher the system usability.

Fifteen Masters and Doctoral students from different departments were randomly selected to evaluate the ATRS system. Figure 15 and Table 1 show the average rating of the all items and four dimensions, and ARTS exhibited high system availability. Each student was asked to itemize the three most useful features of ARTS. The results show the most mentioned advantages included “interesting topic”, “convenient interface”, “easy to use”, or “enables a tourism experience from a remote site”. A few problems were also mentioned, such as “the button layout and interface can be improved” or “the model details and display lag can be improved”.

## 10. Discussion

A full-scale experience is achieved through a 3D time of flight (ToF) scan that creates a 3D point model at a 1:1 scale. Compared to the scale adjustment usually made to a photogrammetry model afterward, a file format conversion to an AR-friendly format is more straightforward. The as-built geometries of the point cloud are converted from OBJ to FBX format. In contrast to the photogrammetry model with high fidelity but with the auto hole-filling of the mesh model, the level of structural detail is greater with scans registered from multiple orientations. The sense of scale enables an accurate perception of subject size, street width, and walking distance with correctly estimated relative dimension by testers.

Scaled interactions between a tester and an enclosed alley or environment are much better, since human configuration or motorcycles are usually scanned accidently as a direct reference of model proportion. The size of the configuration enables the tester to obtain a perfect reference by comparing it to street width or floor height, and creates a familiar recollection of daily visual experience of depth and dimension. The scale can be further perceived by referring to the speed made by the body movement of distance, using real human size as a reference.

### 10.1. Correlating System and Spatial Experience

ARTS exhibits feasibility for use in co-relating cultural dependence on tourism for tourists or students as a pedagogical aid. The scenario-based display for a quiz can provide background block-out, additive or selective display, animated display, pedagogy-based display and interaction, and even can provide learning evaluation records afterwards. The interference of the surrounding physical environments can also be eliminated or selectively displayed only through holes to increase the fidelity of AR scenes. A high-resolution mesh can be displayed as switched. Animated instruction or object movement and rotation can also be added for instructional purposes.

Working with AR inside a space is close to an architecture space walk-through experience and is a much more realistic experience concerning the level of detail. A browsing of a long narrow concave polyhedron is targeted. It is already very common to browse inside an environment, such as a panorama view or indoor VR. However, many AR applications are convex object-based browsing experiences, such as standing outside or next to an object. Not only does the enclosed alley space present a very interesting walking experience, but also the supporting geometry-like sections can be added as an auxiliary interpretation of the varying profiles that a tourist might expect.

### 10.2. The Concerns in Model-Creating Process

The time-evolving monitoring of changes involves the efficiency, accuracy, and scope of related processes. A 4D reconstruction of a scene can be efficiently made, even with moving objects [30,42]. The images taken from different angles should be deployed in advance. Under the 4D scope of a street or a city, a large number of fixed image-taking devices have to be continuously deployed outdoors in public or private properties, in contrast to a simpler setting made in a laboratory. A trip made by a UAV may take 1000–2000 images, and the real-time processing of the images might not be possible. Even if the time-evolving monitoring is made once a month—plus extra time to process images—the trade-off between the efficiency and the area of modeling will need to be evaluated. In addition, the final quality of the 3D models in terms of structural detail and visual detail may vary by the number and availability of images from different angles. A typical concern in creating a model using UAV in the air may come from the missing parts that are blocked by window canopies. Although holes can be filled algorithmically, objects hidden underneath will make the monitored result inaccurate. If additional images have to be taken to recover the blind regions afterward, a registration process has to be made such as that for multiple point clouds.

A long-term 3D scan process is certainly a positive goal for any urban-related study. Although this used to be a time and effort-consuming process, the new development of hardware and software of 3D scanning systems has enabled an automatic registration function of point cloud sets. An additional advantage is the better balance between the accuracy and the scope of the area to be scanned. If a tourism experience is emphasized, a model with rich detail is certainly more attractive to potential tourists. In addition, the full-scaled data can also serve the needs required by architectural history classes or professional practice in planning or remodeling. A 3D scan approach has been very helpful for follow-up studies that need accurate measurements in an urban space.

The 3D scanned alley model of our study recorded a highly detailed configuration with accuracy. The 3D model was overlaid with maps as a way to illustrate the changes made over decades. One of the results is shown to the left of Figure 1. Although our monitoring was not performed for the entire city, hopefully a similar approach can be applied consistently and chronologically in order to construct a solid 3D database as a comparing reference for evolving evaluation. This study analyzed historical information to verify the evolving process in the alley. The illustration was made by deleting or hiding conflict construction to highlight the changes. This is also a kind of 4D representation of the evolving process.

### 10.3. Qualitative and Quantitative Justification

Justifications of the complexity and deformation of the 3D scan approach and related data-feasibility comprise of qualitative and quantitative concerns. The former includes the application of data, image numbers and characters, sensor settings, source of data, correspondence to pedestrians’ viewpoints, model quality, type of subject, items that can be shown (e.g., window frames, landscape), extended application of data, existing software platforms to be interacted, identification of places from façade elements on the ground level, or tourism experiences. The latter includes the scope or the size of the target, the distance between the sensor and the subject, resolution, tolerance, accuracy of the model, and processing time. Pre/post-processing, 3D modeling, and analysis of scanned 3D data may apply [43,44,45]. Both qualitative and quantitative concerns assisted to create a macro-perspective of history and alley fabric, as well as a micro-perspective of sections for purposes of tourism.

Quantitative justification concerns the accuracy of scans, the resolution of decimated models, and the percentage of decimation. The accuracy defined by a 3D scanner reaches 2 mm/30 m, i.e., tolerance is approximately 2 mm in the range of 30 m when a point cloud is created. Accuracy enables a much more detailed description of an object’s configuration, in which a small or detailed one can be seen or identified. Justifications of complexity and deformation can be made at the mm level. In the AR model, the distance between two points under a bridge can be measured as small as 2.2 mm in a real-scale within the decimated cloud model. Polygon numbers and file size post-decimation can still enable a detailed inspection of configuration. Accuracy and resolution are also maintained in a 360 × 300 degree scan, while retrieving almost all of the unobstructed as-built objects in an environment.

The relative location between objects is also maintained. Perfect alignment between the point cloud and the cadastral map provides another type of justification available to inspections made from a collaborated source of legal documents, instead of internal parameters or outcomes from a specific algorithm.

The evaluation of justification does not all necessarily come from the values of tolerance or the efficiency of an algorithm. In this study, we argue that qualitative issues are as critical as quantitative ones. Consequently, the purpose or expected performance of data was projected based on former experience or review of researches. In fact, the time ratio between a field 3D scan and a follow-up data process varies from 1:1 to 1:X. Moreover, the efforts or resources used in 3D scans may exceed people’s perceptions, especially when the data can be reloaded for time-evolving evaluation with historical cadastral maps in the future.

Many local governments’ heritage preservation projects have included 3D point clouds as legal documents. At a survey level of accuracy, the documents are to be archived with the original contract for a period of years. It is interesting that the 3D cloud model can fulfill the needs of real tourism and can be utilized as an accurate legal document as well.

### 10.4. The Notes in Preparing and Using the App

Nevertheless, there are a few aspects that users should be aware of while preparing and using the app. Holes exist in 3D scans due to the obstruction of objects. The alley has been scanned from different directions to cover most of the exposed faces. In order to preserve the original appearance, no automatic hole filling was applied to the mesh model. As a result, the enclosure of a space can be reduced in a brighter background. This problem can be alleviated by projecting the model in a dark room to highlight objects in the foreground. The smartphone camera was also occasionally blocked by fingers. A blocked camera lens can cause a drift problem between virtual objects and real scenes with incorrect relative positioning. Although an approximate estimation of relative location can still be maintained with the assistance of visual-inertial odometry (VIO), drift and unexpected movement reduced the reality of human-smartphone interaction while AR was applied.

## 11. Conclusions

Fabric manipulation constitutes an advanced stage of cloud data conversion. Moreover, the interaction applied to converted data opens up a new domain of manipulation. AR has created a totally open arena of pedagogy and tourism, developing new issues in culture and the environment with experience enabled at a human scale. The issues can be defined in either an abstract form of social value or a solid shape for evaluation, with imagination as the only limitation.

ARTS, an AR tourism system, was developed as a platform using a scanned source of data and popular supporting tool for a captivating interactive experience. The interface is quickly customized for situational adaption. The fabric features a multiple configuration that can be represented in a predefined manner to facilitate specific studies in eTourism and architectural pedagogy. The exploration of as-built data types and interactive forms also greatly broadens the learning environment by taking advantage of the convenience of the smartphone.

A smartphone-based AR learning environment has been proven to be a scalable system with a high level of efficiency and effectiveness. The diversity of remote-sensing data has been demonstrated to be useful for spatial dimension retrieval, thematic space experience, specific pedagogical applications, on-site learning, and remote cultural exploration. The AR display and pedagogical test have led to several findings. Compared to traditional applications of LIDAR data or display, the visual advantage is maintained, in addition to the full-scale dimension of a space. The full-scale experience not only provides tourists with a real sense of being in a space with the correct reference of a human figure, but its combination with colors and textures also makes a space so real that a user can understand and experience the current as-built situation. The integration of structural detail and visual detail facilitates a high level of reality for design, history, and urban-related courses. A pedagogical device can be built based on the mobility and convenience of the smartphone. Annotation data can be designed to meet pedagogical needs by adding a submenu of mediated information to the existing function hierarchy.

A related future goal would be to conduct group browsing or broadcasting at a distance or in different cities by using the app through Wi-Fi or 5G in one-to-one, one-to-many, or many-to-one scenarios. Distributed or remote AR collaboration would be emphasized by streaming data from one iPhone to another at the same time. Although it is feasible to stream to another smartphone nearby, performance is very slow with an obvious lag, compared to the instant response of an AR scene to the movement of the host iPhone. A group of students or tourists (>2 iPhones) could be used to view a subject from different angles, hideouts, or hierarchies during walk-throughs, without carrying head-mounted displays (HMDs) or computers. Group seminars and displays would be enabled at the same time, allowing individual inspection. A point-to-point (P2P) or broadcast AR app with host DB at the individual or cloud-level should be developed to create a better pedagogical environment or to provide efficient assistance to group tourism.

## Figures and Tables

**Figure 1 sensors-19-03725-f001:**
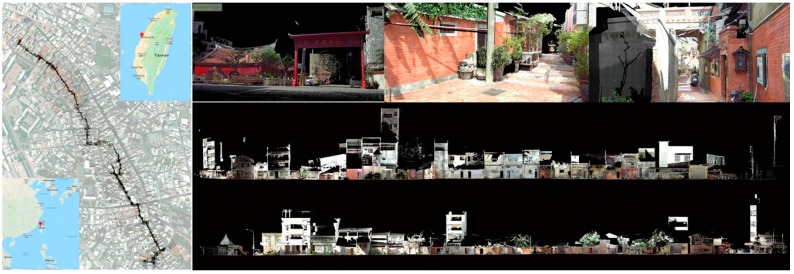
A 2500 m scan of an old street.

**Figure 2 sensors-19-03725-f002:**
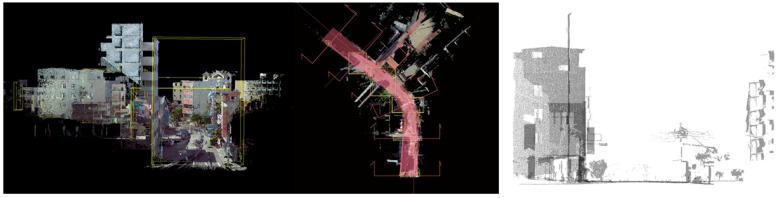
Street block with thick sections sliced along the central line.

**Figure 3 sensors-19-03725-f003:**
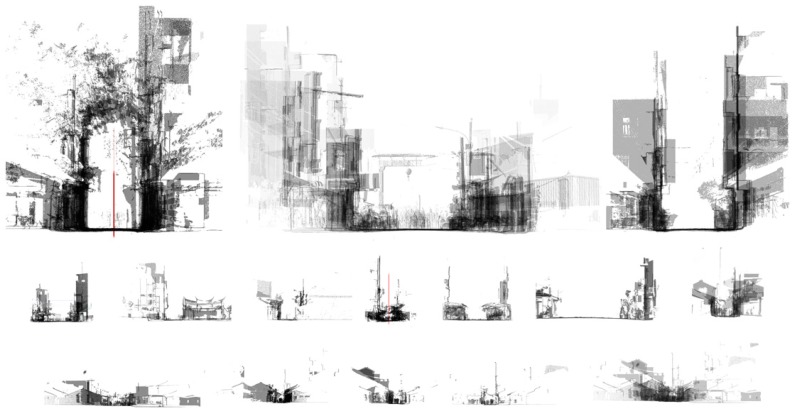
Overlaid sections at every 200 m.

**Figure 4 sensors-19-03725-f004:**
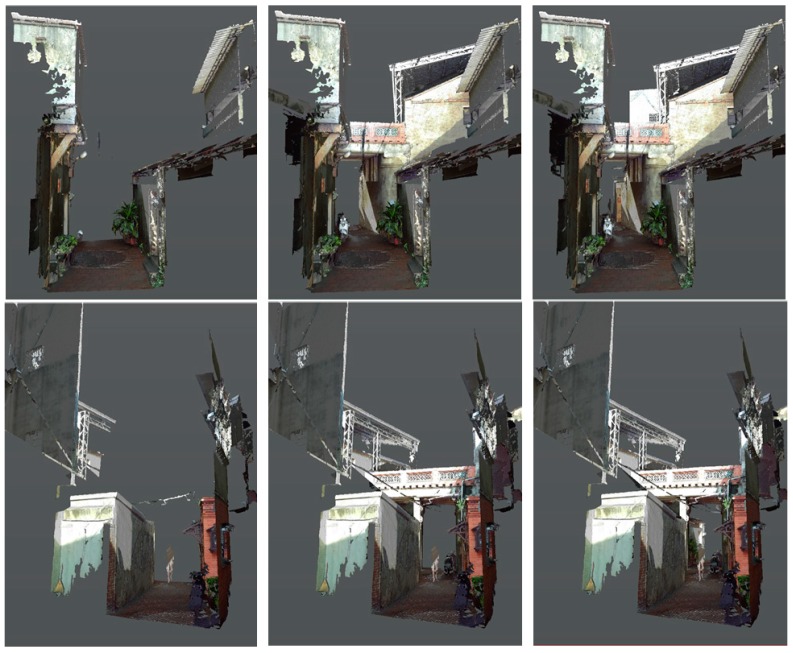
Approaching views of Shih Yih Hall from near to distant (**top**) and from the other direction (**bottom**).

**Figure 5 sensors-19-03725-f005:**
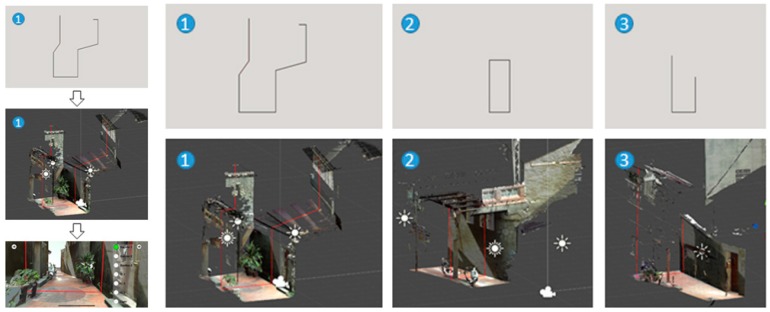
Superimpositions of section annotations (in red line) for an alley in Lukang.

**Figure 6 sensors-19-03725-f006:**
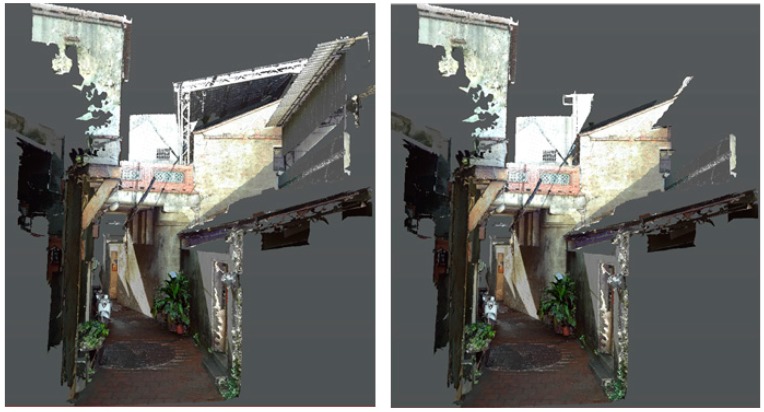
Views of Shih Yih Hall with and without conflict construction removed.

**Figure 7 sensors-19-03725-f007:**
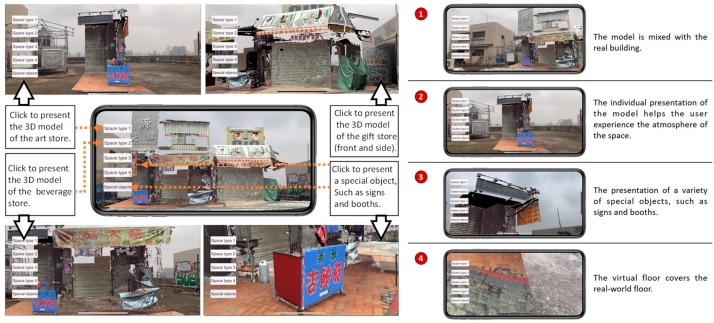
System interface for Yingge.

**Figure 8 sensors-19-03725-f008:**
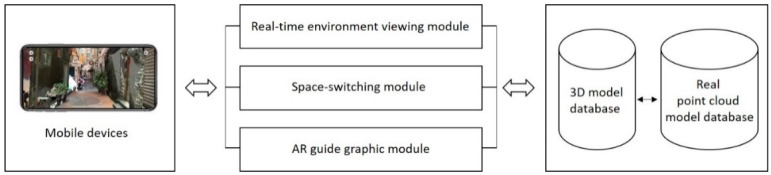
System structure.

**Figure 9 sensors-19-03725-f009:**
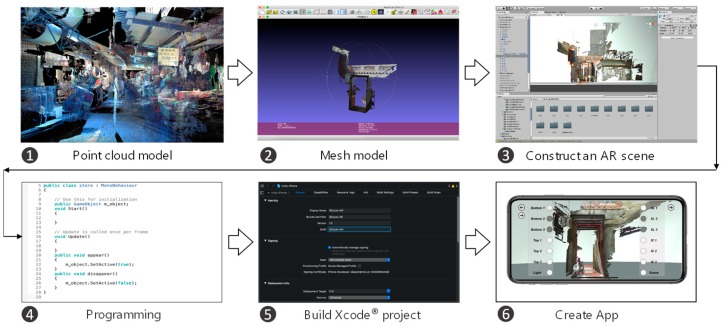
System developing process.

**Figure 10 sensors-19-03725-f010:**
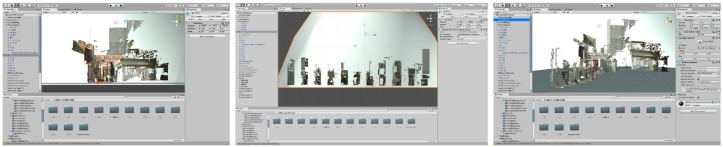
Unity operation interfaces (as shown in 11 slices of the section).

**Figure 11 sensors-19-03725-f011:**
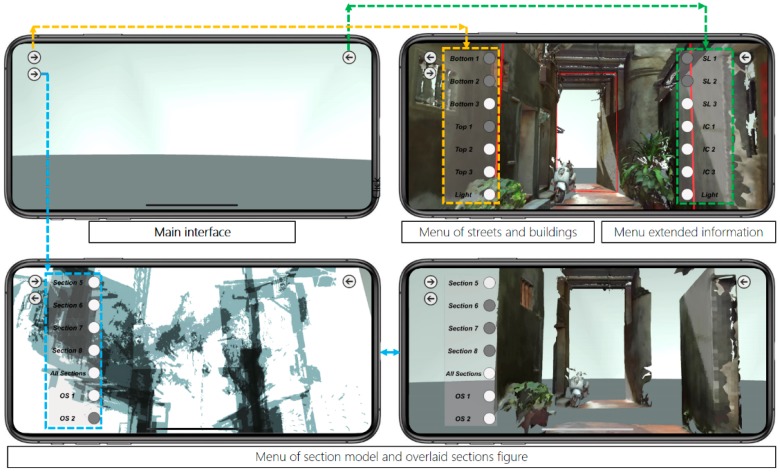
System interface screenshots for an alley in Lukang.

**Figure 12 sensors-19-03725-f012:**
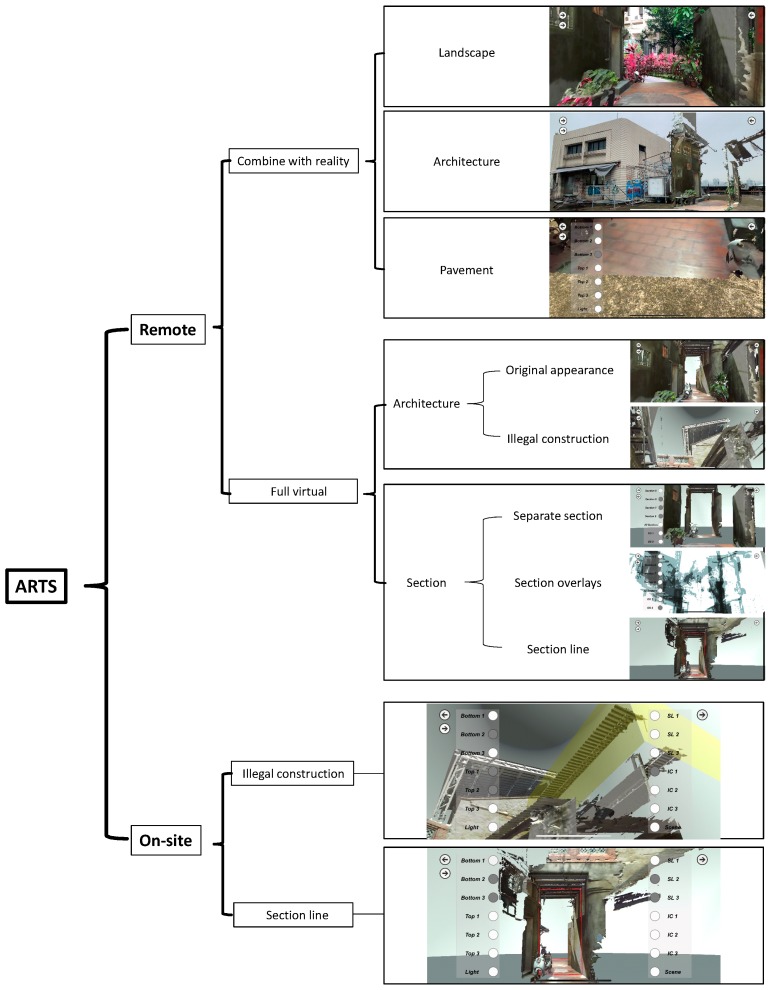
Augmented Reality tourism system (ARTS) system function structure diagram.

**Figure 13 sensors-19-03725-f013:**
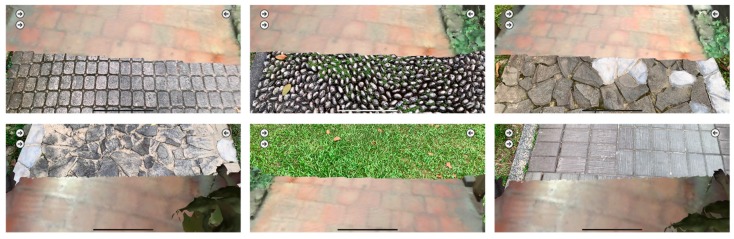
The combination of real paving and 3D models.

**Figure 14 sensors-19-03725-f014:**
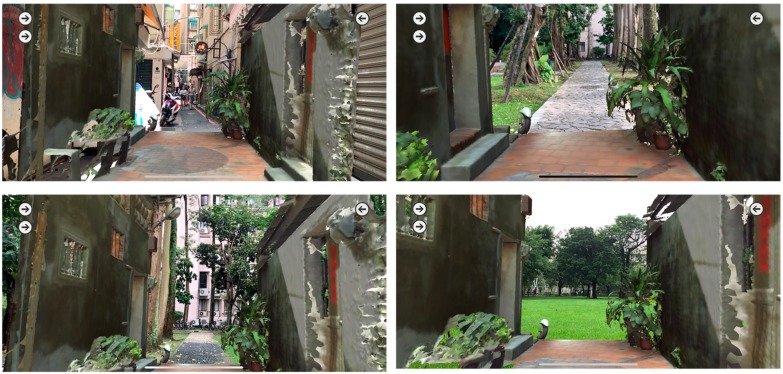
The combination of the real environment and 3D models.

**Figure 15 sensors-19-03725-f015:**
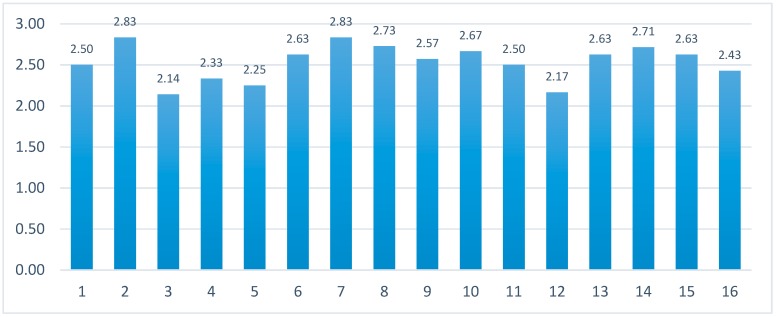
The average score for all items in PSSUQ.

**Table 1 sensors-19-03725-t001:** Average score of the PSSUQ in four dimensions.

Dimensions	Items	Average
Overall average	1–16	2.31
System usefulness	1–6	2.35
Information quality	7–12	2.21
Interface quality	13–15	2.26

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
