# Peer review of "ARTS, an AR Tourism System, for the Integration of 3D Scanning and Smartphone AR in Cultural Heritage Tourism and Pedagogy"

_sensors, 2019, doi:10.3390/s19173725_

Round 1
Reviewer 1 Report
The paper describes the development of an AR tool for the visualization and interaction with 3D data generated both with active and passive sensors.
The structure of the paper is not clear and need a major rearrangement. The reader is not able to grasp the core idea of the paper which is briefly described in the abstract. Many concepts are repeated multiple times but in general, all the article doesn't read well.
From section 1 to 7 the paper is very confusing and different domains ( such as architecture, 3D modelling and pedagogy) are described and juxtaposed without any clear connection between them. These sections might be summarized briefly before section 8 (the description of the system).
Also, the discussion end conclusion sections are not very clear and well elaborated.
To date, the paper doesn't sound mature enough for publication. Due to the availability of such systems (as highlighted in the state of the art section), I would suggest to further develop the user-experience section and, and after a proper assessment of the results, resubmit the paper when more data will be available.
Although I'm not an English native speaker myself, the English level is not clear. besides minor mistakes, many sentences are too wordy and confusing for the reader.
Author Response
Reviewer 1:
l The paper describes the development of an AR tool for the visualization and interaction with 3D data generated both with active and passive sensors.
l The structure of the paper is not clear and need a major rearrangement. The reader is not able to grasp the core idea of the paper which is briefly described in the abstract. Many concepts are repeated multiple times but in general, all the article doesn't read well.
Response: Your comments are highly appreciated. The structure of the paper is re-interpreted with more matching meaning of title for each section. Sections are listed as follows.
1. Introduction
2. Purposes
3. Related work
4. The macro perspective in history and fabric
5. The micro perspective of sections
6. Involved roles
7. AR-related tasks and features
8. ARTS
9. System evaluation
10. Discussions
11. Conclusions
The macro perspective in history and fabric, the micro perspective of sections, and the involved roles were stated, prior to the introduction of AR-related tasks and features and ARTS. A new PSSUQ evaluation was added. The core idea of the paper is briefly described in the abstract.
l From section 1 to 7 the paper is very confusing and different domains (such as architecture, 3D modelling and pedagogy) are described and juxtaposed without any clear connection between them. These sections might be summarized briefly before section 8 (the description of the system).
Response: Architecture study is usually related to the 3D configuration of building. The technology of 3D modeling has to be applied to fulfill this purpose. While the 3D contents are to be used in architectural education, well-prepared pedagogy contents usually include 3D models in order to deliver a clear illustration of visual information. So is the heritage when a study involves 3D artifacts to be displayed to students or potential tourists in a remote site.
Since ARTS involves heritage-related subjects and background, the manuscript is now developed from the knowledge in elate studies, macro and micro perspective defined in Section 4 and 5, roles, to the system and evaluation. Hope the rearrangement and reinterpretation can connect the structure well.
l Also, the discussion end conclusion sections are not very clear and well elaborated.
Response: As discussed in this section 10, several issues were brought up.
n A full-scale experience is achieved through a 3D ToF scan that creates a 3D point model at a 1:1 scale.
n Scaled interactions between a tester and an enclosed alley or environment become much better by referred proportion of objects.
n ARTS is feasible in co-relating cultural dependence in tourism for tourists or students as a pedagogical aid.
n Working with AR inside a space is closer to an architecture space walk-through experience, and is a much more realistic experience concerning the level of detail.
n A few notes, which should be aware of while preparing and using the app, are also listed.
l To date, the paper doesn't sound mature enough for publication. Due to the availability of such systems (as highlighted in the state of the art section), I would suggest to further develop the user-experience section and, and after a proper assessment of the results, resubmit the paper when more data will be available.
Response: Hope the revised manuscript can present the system well. A user-experience section, PSSUQ evaluation, is added and the response of the system is also stated.
l Although I'm not an English native speaker myself, the English level is not clear. besides minor mistakes, many sentences are too wordy and confusing for the reader.
l Response: The manuscript is revised in terms of section titles, newly added text and shorted sentences (in red).
Reviewer 2 Report
This paper provides a augmented reality framework for tourism application scenario. The scheme involves 3D scanning procedures and computer science technology.
The idea is interesting. However, there are some issues to be resolved before the publication of this work.
3D scanning is an expensive and time consuming process. It requires photogrammetric techniques and then processing of the obtained images. However, our words is not static. It is evolving though time resulting therefore into 4D maps. The authors fails to discuss these issues, that is the time evolving 3D scanning. in the literature, there are papers dealing with 3D digitisation. Example includes the work of [1] where a 4D digitisation framework is proposed from wild images, or the work of [2] where 4D reconstruction algorithms are incorporated or even the work of [3]. So, the authors should provide discussions of how the recent methodologies in 4D reconstruction assist the proposed AR application for touristic purposes.
The role of computational complexity is not clearly discussed in the proposed paper. 3D models imposes challenges in memory and processing. So, what are the proposed methods adopted for handling these issues?
In addition, the authors should better describe the biography and include the respective refers in the field of 3D reconstruction for cultural tourism applications (see [4]).
Refs.
[1]. Ioannides, Marinos, et al. "Online 4D reconstruction using multi-images available under Open Access." ISPRS Annals of the Photogrammetry, Remote Sensing and Saptial Information Sciences, II-5 W 1 (2013): 169-174.
[2] A. Mustafa, H. Kim, J. Guillemaut and A. Hilton, "Temporally Coherent 4D Reconstruction of Complex Dynamic Scenes," 2016 IEEE Conference on Computer Vision and Pattern Recognition (CVPR), Las Vegas, NV, 2016, pp. 4660-4669.
[3]A. Mustafa, "General 4D Dynamic Scene Reconstruction from Multiple View Video" PhD thesis, University of Surrey, 2016.
[4]
Shao, Z., Li, C., Zhong, S., Liu, B., Jiang, H., Wen, X.
3D RECONSTRUCTION OF CULTURAL TOURISM ATTRACTIONS FROM INDOOR TO OUTDOOR BASED ON PORTABLE FOUR-CAMERA STEREO VISION SYSTEM
(2015) International Archives of the Photogrammetry, Remote Sensing and Spatial Information Sciences - ISPRS Archives, 40 (4W5), pp. 193-197.
Author Response
Reviewer 2:
l This paper provides a augmented reality framework for tourism application scenario. The scheme involves 3D scanning procedures and computer science technology.
l The idea is interesting. However, there are some issues to be resolved before the publication of this work.
Response: Related issues are stated in the following sections.
l 3D scanning is an expensive and time consuming process. It requires photogrammetric techniques and then processing of the obtained images. However, our words is not static. It is evolving though time resulting therefore into 4D maps. The authors fails to discuss these issues, that is the time evolving 3D scanning. in the literature, there are papers dealing with 3D digitisation. Example includes the work of [1] where a 4D digitisation framework is proposed from wild images, or the work of [2] where 4D reconstruction algorithms are incorporated or even the work of [3]. So, the authors should provide discussions of how the recent methodologies in 4D reconstruction assist the proposed AR application for touristic purposes.
Response: Your comments are highly appreciated. Although 3D scan is a time- or effort-consuming process, the data are very accurate and useful for any possible follow-up inspection. The advantage of 3D scan and its comparison to photogrammetry or 4D reconstruction are included in the manuscript.
n In section 3. Related work: The study of a heritage site usually needs to create a 3D model for preservation or tourism purpose. The model can be used for documentation, measurement, or illustration. The recorded contents not only illustrate the open space, but also the urban fabric that is made of buildings, culture entities, or landscape. Photogrammetry technology has been widely applied in UAV (unmanned aerial vehicle) or UAS (unmanned aerial system) [28-29]. 4D dynamic scene reconstruction has been developed as an efficient modeling approach using videos or photos as input resource of images [30]. For the capture of detailed models, 3D scan can create objects of original scale (in 1:1) to fulfill measurement need in architectural professional practice. An automatic scan process can be made to a scene up to 360*300 degrees of an environment in a range of hundred meters. Automatic registration of multiple scans can be made, no matter it is a large site or a long alley. Point cloud model has been widely used as construction documents.
n In section 10. Discussions: A full-scale experience is achieved through a 3D ToF (Time of Flight) scan that creates a 3D point model at a 1:1 scale. Compared to the scale adjustment usually made to a photogrammetry model afterward, a file format conversion to an AR-friendly format is more straightforward. The as-built geometries of the point cloud are converted from OBJ to FBX format. In contrast to the photogrammetry model with high fidelity but auto hole-filling of the mesh model, the level of structural details is greater with scans registered from multiple orientations. The sense of scale enables an accurate perception of subject size, street width, and walking distance with correctly estimated relative dimension by testers.
l The role of computational complexity is not clearly discussed in the proposed paper. 3D models imposes challenges in memory and processing. So, what are the proposed methods adopted for handling these issues?
Response: The size of 3D scan data was decimated by the number of points or polygons. The tradeoff is, one the one hand, to keep as much details as possible, and on the other hand, reduce the computation or display burden of the mobile phone. So an empirical experiment was conducted by separating a big model by parts and resolutions and imported to the phone one by one to see if the system still works. We actually did three versions of decimation percentage and ended up with the current feasible combinations. Hopefully in the future, the concerns of memory size and processing capacity can be alleviated, or 5G and cloud computing can reduce the computation burden of a mobile phone.
The contents mentioned in the manuscript are listed as follows.
n In section 8.3. The data work flow and scan data conversion: Due to the limited storage space of a mobile phone, an alley approximately 20 m next to Shih Yih Hall was selected. The representation and visualization is divided into two categories for the approaching views using connected six parts (Figure 6) and the profile views using overlaid sections of 11 separated slices (Figure 10).
n Due to the limited storage space of a mobile phone, an alley approximately 20 m next to Shih Yih Hall was selected. Increasing polygon number causes a delay in the screen display with flickers. Decimation was made by defined polygon count or percentage. The entire model is made around Shih Yih Hall. For display efficiency, the original mesh model of 301.4 MB was divided into 18 parts in six adjacent facades in a top and bottom arrangement, 11 façade slices along the street axis, and one overlaid façade of the 11 slices. The first larger model was divided into six parts. Each part was approximately 400000 - 1000000 polygons, comprising about 32% of the original polygon numbers. The six sizes of all six parts were reduced from 668.7 MB in PLY format, to 324 MB in OBJ format, to 159.1 MB in FBX format, constituting 22.5% of the original size. The 11 slices, which was made of 60 million points in about 1 GB, were used to illustrate the alley. The data were decimated and converted to about 2.2 million polygons in 452 MB, prior to the final size of 1000000 polygons were achieved. The 11 parts were also combined. Both separated and combined models 142.3 MB.
l In addition, the authors should better describe the biography and include the respective refers in the field of 3D reconstruction for cultural tourism applications (see [4]).
Response: 4D reconstruction of dynamic scenes is newly added and discussed in the manuscript. 4D dynamic scene reconstruction has been developed as an efficient modeling approach using videos or photos as input resource of images [30]. A related reference is added. Two new photogrammetry-related references [28-29] are also added.
n Nex, F.; Remondino, F. UAV for 3D mapping applications: A review. Applied Geomatics 2014, 6, 1-15. doi: 10.1007/s12518-013-0120-x.
n Colomina, I.; Molina, P. Unmanned aerial systems for photogrammetry and remote sensing: A review. ISPRS Journal of Photogrammetry and Remote Sensing 2014, 92, 79-97. doi: 10.1016/j.isprsjprs.2014.02.013.
n Mustafa, A.; Kim, H.; Guillemaut, J.Y.; Hilton, A. Temporally coherent 4D reconstruction of complex dynamic scenes, In Proceedings of the IEEE Computer Society Conference on Computer Vision and Pattern Recognition, CVSSP, University of Surrey, Guildford, UK, 26 June -1 July 2016; pp 4660-4669.
Round 2
Reviewer 1 Report
In the revised version of the paper, the authors have improved the readability and the overall structure of the article. Now there is a much clearer flow of information throughout.
Please clarify better the use of photogrammetry in the workflow for the creation of the 3D model. Was it used for the creation of DEM? Was it used to complete the upper side (roofs) of the buildings? Was it used to integrate the missing areas impossible to collect from the ground level?
I would suggest clarifying the use of sections of the 3D point cloud since it might not be clear for the reader to grasp their use inside the AR environment. This applies also for the potential tourists or non-specialist users using the App.
The images used in the section concerning the evaluation of the APP are not properly intelligible. A scale with the different values, as noted in the text, would help the reader to understand the outcomes better.
The clarity of English now is improved, although typos are still present.
Author Response
Dear Reviewer 1:
On behalf of my co-authors, I would like to submit the response to your comments. The texts in blue were edited and added to the revised Section 5, 8, 8.2, and 9.
Your assistance is appreciated.
Best regards,
Naai-Jung Shih

Reviewer 2 Report
The authors fails to provide adequate justifications about the complexity of this work. particularly, they said "Your comments are highly appreciated. Although 3D scan is a time- or effort-consuming process, the data are very accurate and useful for any possible follow-up inspection. The advantage of 3D scan and its comparison to photogrammetry or 4D reconstruction are included in the manuscript." This is the issue. A precise 3D reconstruction requires many man power (high cost, including human resources. So, this approach is not practically implementable. In order to address this issue, we need to include selective partial 3D modelling. This is the main research results of the 4DCHWORD EU funded project, aiming to provide 4D reconstruction solutions of dynamic 3D environments like the ones that this paper contribute to. The idea is to reconstruct only regions of significant change. In this way a partial 3D mapping is encounters (see [1]). This should be clearly discussed in the revised manuscript. Otherwise the proposed AR approach is static and within a time period is obsolete. This is critical for a practical implementation of this research.
[1] A. Doulamis, S. Soile, N. Doulamis, C. Chrisouli, N. Grammalidis, K. Dimitropoulos, C. Manesis, C. Potsiou, C. Ioannidis, “Selective 4D modelling framework for spatial-temporal land information management system,” Proceedings of SPIE - The International Society for Optical Engineering, 9535, Paphos, Cyprus, 2015.
Author Response
Dear Reviewer 2:
On behalf of my co-authors, I would like to submit the response to your comments. The texts in blue were edited and added to the revised Section 10.2 and 10.3.
Your assistance is appreciated.
Best regards,
Naai-Jung Shih

Round 3
Reviewer 1 Report
The authors have addressed sufficiently the critical sections of the manuscript.
I would suggest having the paper revised by the MDPI service (as stated in point 4 of authors comments) before publishing.
Author Response
Dear Reviewer:
On behalf of my co-authors, the manuscript was revised and a few errors were corrected. However, the revision was asked to be returned in one day, which might not be long enough for a contract-out revision.
The third revision was mainly related to the relevant references added to the revised part. It was considered that both the two revisions (the 1st and 2nd rounds) were made thoroughly by native English speakers. Unfortunately, it seems still there were errors undiscovered from the two revisions. I’m really sorry for this. The 2nd round of manuscript revision was made of two parts: the new text 1 and the new text 2 that was added after the text 1 was sent to MDPI editing service. Since the editing system did not allow adding another part of the manuscript afterward, the new text 2 was then sent to a native English speaker who also edited the entire manuscript at the round 1. Both certificate and receipt were attached to earlier cover letters.
I have contacted the editor regarding this limited time span in editing and was told that there will be another MDPI service including layout and English editing afterward. Of course, this is feasible only if the manuscript is accepted. Hopefully new editing service will be applied.
Your comments are highly appreciated.
Best regards,
Naai-Jung Shih
Reviewer 2 Report
The revised section of 4D modelling should be justified with relevant references.
Author Response
Dear Reviewer:
On behalf of my co-authors, an existing reference of [30] was added to Line 457, followed by a new reference [42], in the sentence regarding the 4D reconstruction of a scene in the “Section 10.2. The concerns in model-creating process.” It is where the 4D justification was mentioned in the “Section 10. Discussions”, other than the “Section 3. Related work.” This sentence stated that: “A 4D reconstruction of a scene can be efficiently made, even with moving objects [30,42].”
Three new references were added at Line 495 regarding the post-processing of scanned 3D data in the “Section 10.3. Qualitative and quantitative justification”, where the 3D scan was justified. This sentence stated that: “Pre/post-processing, 3D modeling, and analysis of scanned 3D data may apply [43-45].”
More detailed descriptions can be found in the file attached.
Your comments are highly appreciated.
Best regards,
Naai-Jung Shih
